# Past Connectivity but Recent Inbreeding in Cross River Gorillas Determined Using Whole Genomes from Single Hairs

**DOI:** 10.3390/genes14030743

**Published:** 2023-03-18

**Authors:** Marina Alvarez-Estape, Harvinder Pawar, Claudia Fontsere, Amber E. Trujillo, Jessica L. Gunson, Richard A. Bergl, Magdalena Bermejo, Joshua M. Linder, Kelley McFarland, John F. Oates, Jacqueline L. Sunderland-Groves, Joseph Orkin, James P. Higham, Karine A. Viaud-Martinez, Esther Lizano, Tomas Marques-Bonet

**Affiliations:** 1Institute of Evolutionary Biology (UPF-CSIC), PRBB, c/ del Dr. Aiguader 88, 08003 Barcelona, Spain; 2Center for Evolutionary Hologenomics, The Globe Institute, University of Copenhagen, Øster Farimagsgade 5A, 1352 Copenhagen, Denmark; 3Department of Anthropology, New York University, New York, NY 10003, USA; 4New York Consortium in Evolutionary Primatology, New York, NY 10065, USA; 5Conservation, Education and Science Department, North Carolina Zoo, Asheboro, NC 27205, USA; 6SPAC Scientific Field Station Network, Hasso Plattner Foundation (HPF), 14467 Potsdam, Germany; 7Department of Ecology and Environmental Sciences, University of Barcelona, 08028 Barcelona, Spain; 8Department of Anthropology, James Madison University, Harrisonburg, VA 22807, USA; 9Triton College, River Grove, IL 60171, USA; 10Department of Anthropology, Hunter College, City University of New York, New York, NY 10065, USA; 11Department of Forest Resources Management, University of British Columbia, Vancouver, BC V6T 1Z4, Canada; 12Department of Anthropology, Montreal University, Montreal, QC H3T 1N8, Canada; 13Illumina Laboratory Services, Illumina Inc., 5200 Illumina Way, San Diego, CA 92122, USA; 14Institut Català de Paleontologia Miquel Crusafont, Universitat Autònoma de Barcelona, Edifici ICTA-ICP, Cerdanyola del Vallès, 08193 Barcelona, Spain; 15Catalan Institution of Research and Advanced Studies (ICREA), Passeig de Lluís Companys, 23, 08010 Barcelona, Spain; 16CNAG-CRG, Centre for Genomic Regulation (CRG), Barcelona Institute of Science and Technology, Baldiri i Reixac 4, 08028 Barcelona, Spain

**Keywords:** non-invasive, hairs, inbreeding, bottleneck, gene flow, wild gorillas, NGS, Cross River gorilla

## Abstract

The critically endangered western gorillas (*Gorilla gorilla*) are divided into two subspecies: the western lowland (*G. g. gorilla*) and the Cross River (*G. g. diehli*) gorilla. Given the difficulty in sampling wild great ape populations and the small estimated size of the Cross River gorilla population, only one whole genome of a Cross River gorilla has been sequenced to date, hindering the study of this subspecies at the population level. In this study, we expand the number of whole genomes available for wild western gorillas, generating 41 new genomes (25 belonging to Cross River gorillas) using single shed hairs collected from gorilla nests. By combining these genomes with publicly available wild gorilla genomes, we confirm that Cross River gorillas form three population clusters. We also found little variation in genome-wide heterozygosity among them. Our analyses reveal long runs of homozygosity (>10 Mb), indicating recent inbreeding in Cross River gorillas. This is similar to that seen in mountain gorillas but with a much more recent bottleneck. We also detect past gene flow between two Cross River sites, Afi Mountain Wildlife Sanctuary and the Mbe Mountains. Furthermore, we observe past allele sharing between Cross River gorillas and the northern western lowland gorilla sites, as well as with the eastern gorilla species. This is the first study using single shed hairs from a wild species for whole genome sequencing to date. Taken together, our results highlight the importance of implementing conservation measures to increase connectivity among Cross River gorilla sites.

## 1. Introduction

Equatorial Africa is home to two species of gorilla, which are distributed across a distance of 900 km and are the subjects of major conservation concern. The vast majority of Western gorillas belong to the western lowland gorilla subspecies (WL, *G. g. gorilla*), which has the largest distribution range and census population size of all gorilla subspecies (362,000 individuals) [1]. In contrast, Cross River gorillas (CR, *G. g. diehli*), the other subspecies of Western gorilla, exist in geographically distinct subpopulations [2,3] across the border between Nigeria and Cameroon and have the smallest census population size (~250–300 gorillas) [4,5]. Eastern gorillas are split into Grauer’s gorillas (GR, *Gorilla beringei graueri*), which have the second largest distribution range and census population size (6800 individuals) [6], and mountain gorillas (MN, *Gorilla beringei beringei*), which live in two discrete regions, the Virunga Massif and Bwindi Impenetrable National Park. The MN census population size has increased in recent years (above 1000 individuals) [7] due to successful conservation efforts. With the exception of MN gorillas (considered endangered) [8], all other gorilla subspecies are listed as critically endangered in the IUCN Red List of Threatened Species [9,10,11].

Anthropogenic factors such as habitat loss and degradation and former hunting activities have impacted CR gorillas, confining them to refuge areas to avoid human pressure [12,13,14,15]. We expect this increase in habitat fragmentation to hinder connectivity among CR gorilla populations and lead to high inbreeding levels, which, together with drift, could lead to reduced genetic diversity.

Despite great effort, the present understanding of CR gorilla population genomics has been limited by the small number of high-quality samples available for whole-genome sequencing. Most studies have relied on microsatellites [14,16,17,18,19,20], and to date, only one individual CR genome has been sequenced [21]. These results indicated that the heterozygosity level of CR gorillas falls in between that of WL and GR gorillas [14,20,21]. The divergence between CR and WL from their common ancestor is estimated to have occurred 17–114 thousand years ago (kya), although gene flow continued until ~420 years ago (ya) [21,22,23]. Furthermore, it has been suggested that CR gorillas have suffered a recent reduction in population size with a strong bottleneck around 100–200 ya [14,22]. Fecal microsatellite studies have identified three primary subpopulations of CR gorillas that coincide with patterns of habitat fragmentation [19], with occasional migration between them [4,19]. Given the limitations of the available data, assessing present levels of genetic diversity, inbreeding, population substructure, and levels of gene flow among subpopulations will be important not only for the conservation of CR gorillas but also for understanding their evolutionary history.

In this study, we generated whole-genome data from 115 single hairs collected across the western gorilla range, comprising the largest dataset of CR gorilla whole genomes sequenced to date. We use this dataset to deepen our understanding of the demographic history and connectivity of CR gorillas. As suggested by previous studies, we uncover recent inbreeding among CR gorillas using runs of homozygosity. However, we also confirm the evidence of past gene flow both among CR subpopulations and between CR gorillas and the other gorilla subspecies. Taken together, these findings improve our understanding of the genetic diversity and evolutionary history of CR gorillas and highlight the urgent need to implement conservation measures for the Cross River gorillas. 

## 2. Materials and Methods

### 2.1. Samples, DNA Extraction, and Library Preparation

A total of 115 single shed hairs from putative gorillas were used mainly from an archived collection in the Primatology Department at NYU. These were mostly collected between 1991 and the early 2000s from CR and WL gorilla sites and three unknown sites (Appendix A). 

DNA was extracted from previously cleaned individual hairs using the QIAamp DNA Investigator Kit (QIAGEN, Cat. No./ID: 56504) following the manufacturer’s instructions. A total of 105 sequencing libraries were made in Claret Bioscience (Santa Cruz, CA, USA), with 18 µL of DNA extract following the Santa Cruz Reaction (SCR) protocol [24] modified for small fragment retention. The remaining libraries (*n* = 11) were prepared using 20 µL of DNA extract following the SCR protocol [24] at the Comparative Genomics Group (IBE-UPF, Barcelona, Spain). For one sample, two libraries were generated, resulting in a total of 116 libraries from 115 hairs. Libraries were sequenced on an Illumina NovaSeq 6000 with a 2 × 150 bp setup at the National Center for Genomic Analysis (CNAG, Barcelona, Spain).

### 2.2. Sequencing Data Processing (FASTQ to Genotypes/Likelihoods)

Sequencing reads were trimmed, quality filtered, and merged with FASTP (version 0.23.1) [25]. Mapping to the hg38 reference genome (GenBank GCA_000001405.15) of merged and paired reads (not merged) was performed with the bwa mem algorithm (version 0.7.15) [26], and read groups were added with PICARD (version 1.95) [27]. To avoid spurious alignments, mapped reads with read lengths below or equal to 35 bp were removed (as suggested by Meyer et al. [28] and Kuhlwilm et al. [29]). All bam files corresponding to the same library were merged into one with samtools “merge” (version 0.7.15) [30]. Each bam file was further processed to remove duplicates with PICARD “markduplicates” (version 1.95) [27] and filtered with samtools (version 0.7.15) [30]. 

FASTQs from high-coverage sequencing of wild gorillas published in previous studies (HighCov samples from now on) [21,31] were mapped to the human reference (hg38) following the same pipeline as the hair samples. Several of these high-coverage published samples were downsampled to an average coverage of 0.75×, 1×, 3×, 7×, and 10× to be used in further analyses (Appendix A).

### 2.3. Quality Control (Depth of Coverage and Contamination)

The average coverage of each library was estimated with MOSDEPTH (version 0.2.6) [32]. Two different approaches were followed to check for potential contamination of the hair samples. The Human Contamination Test for Great Ape Samples (HuConTest) script [29] was used, first setting ‘gorilla’ and then ‘pan’ (chimpanzee or bonobo) as the species to test. This script allowed us to obtain an estimation of the percentage of putative human contamination of the samples, as well as to confirm the presence of several chimpanzee samples that were misidentified as gorilla hairs (Appendix A). The BBsplit software (BBMAP 38.57) [33] was used on those samples with HuConTest results above 5% (>5% human contaminated hairs) as a secondary approach to confirm HuConTest results. BBsplit consists of a competitive mapping strategy where each read is aligned to a range of different genome assemblies from other primates. Apart from the human and the gorilla reference genomes (GRCh38 and gorGor6, respectively), reference genomes corresponding to primate species with overlapping distribution ranges to gorillas were used: chimpanzee (panTro6), De Brazza’s monkey (CertNeg_v1_BIUU), Angolan colobus (Cang.pa_1.0), olive baboon (Panu_3.0), mandrill (mandrill_1.0), and drill (Mleu.le_1.0) (Appendix A). We considered a sample to pass the quality control (QC) if it had an estimation below 1% of putative human contamination from the HuConTest and coverage ≥0.5× (Appendix A).

### 2.4. Genotype Likelihoods and SNP Calling

Genotype likelihoods for the whole set of samples and several subsets were computed using ANGSD (version 0.931) [34]. We chose the SAMtools model as the method to estimate genotype likelihoods and restricted the computation to only autosomal chromosomes. In each set of samples, we specified that at least 80% of individuals in each variant site needed to be present. Furthermore, we applied several quality filters: “-uniqueOnly 1 -remove_bads 1 -only_proper_pairs 1 -trim 0 -C 50 -minMapQ 30 -minQ 20 -SNP_pval 1e-6”. We specified to output the genotype likelihoods in nonbinary beagle format. These genotype likelihoods were used for PCA and admixture analyses.

The whole set of samples (Set01) consisted of all hair samples before QC together with the HighCov samples (see Section 2.3); thus, it contains samples from all four gorilla subspecies. Set02 was generated from Set01, keeping those samples that passed QC (HuConTest <1% and average coverage ≥ 0.5×) and that were kept after relatedness analysis (see Relatedness section). Set03 and Set04 derive from Set02, keeping WL gorillas and CR gorillas, respectively. Finally, Set05, Set06, and Set07 derive from Set02, keeping western gorilla hair samples, WL hair samples, and CR hair samples, respectively (Appendix A).

Genotypes were called for those hair samples from Set02 with coverage above 7× together with the high-coverage reference samples [21,31] (HighCov dataset; Appendix A) using GATK Haplotype Caller (version 4.1.4.0) [35,36]. Genotyping was performed for these samples with GenotypeGVCFs from the GATK toolkit. The resulting VCF was filtered with GATK SelectVariants to keep only SNPs and further filtered following GATK hard-filtering best-practice guidelines. Furthermore, we used BCFTools Filter (version 1.9) [30] to keep only those positions with an average coverage between a third and three times the dataset’s average coverage. We filtered those heterozygous positions with an allele balance <0.15 or >0.85 with BCFTools (version 1.9) [30] to discard any potential residual contamination. We also generated a dataset of these high-coverage samples to an average coverage of 10x (Down10× dataset).

### 2.5. Genetic Sexing from Coverage

For each library, the coverage per chromosome was estimated with MOSDEPTH (version 0.2.6) [32], and from this, the average autosomal coverage was obtained. Then, we estimated the ratio between the X chromosome coverage and the average autosomal coverage. We assigned samples as belonging to a female, male or undetermined following Cabrera et al. [37] (Appendix A).

### 2.6. Relatedness Test

We ran NgsRelate (version 2) [38] per gorilla site (Appendix A) on samples passing the QC. Where first- or second-degree relatives were detected, one individual from each pair was excluded from population structure analyses. From each related pair, we chose to exclude the sample with the lowest coverage and the higher number of familial relationships in the dataset. 

### 2.7. Population Structure Assessment

We obtained beagle files of the different sets of samples (described in Genotype likelihoods and SNP calling section; Appendix A) to be used as input for both PCAngsd (version 0.8) [39] and for NgsAdmix (ANGSD version 0.916) [40]. We ran NgsAdmix for multiple Ks and evaluated its results using EvalAdmix [41] (Appendix A).

Pairwise Fst estimates between different sites were calculated with ANGSD [34]. We used three randomly selected samples per site when more were available (Afi Mountain, Mbe, Kagwene, Mbe, DengDeng, MonteAlen, and NgagaCamp) (Appendix A). Bai Hokou and Lobéké sites were not included in this analysis because we only had one sample for each after QC.

To calculate pairwise Fst, we first obtained site allele frequency likelihood (saf) files for each population as follows: “angsd -b $file -ref Hg38.fa -anc Hg38.fa -sites autosomes.file -dosaf 1 -gl 1 -uniqueOnly 1 -remove_bads 1 -only_proper_pairs 1 -trim 0 -C 50 -baq 1 -minMapQ 20 -minQ 20”. Then, we calculated the 2D site frequency spectrum (SFS) with realSFS. This output was used as priors together with the saf files from the first step, obtaining fst binary files with realSFS “fst index” setting “-whichFst 1”. This setting is preferable for small sample sizes such as our case. Finally, realSFS “fst stats” was used to extract the Fst values, which were plotted with R (version 4.1.2) [42].

### 2.8. Heterozygosity Estimates

We estimated genome-wide heterozygosity per autosomal chromosome for the hair samples that passed our QC (HuConTest < 1% and average coverage ≥0.5×) and for the HighCov samples (full coverage and downsampled several coverage ranges) using ANGSD (version 0.931) [34].

To account for differences in coverage in our dataset, we downsampled hair samples and HighCov samples with coverage >1.1× to 1× with Picard (version 2.20.0) [27] (Appendix A). Then, we estimated the genome-wide heterozygosity of the resulting dataset of coverage ≤1× (downsampled hairs and published data with hairs <1× coverage) (Appendix A). 

### 2.9. Estimation of Runs of Homozygosity and Inbreeding Coefficients

To estimate runs of homozygosity (ROH) along the genome, we used BCFTools (version 1.14) [43] on the Down10x dataset (see Appendix A). We used this dataset to avoid biases in the estimation of runs of homozygosity from differences in coverage between samples. We used a recombination rate of 1 × 10^−8^ [31] and kept only RoH with size ≥50 kbp and quality ≥70. To estimate heterozygosity by windows on the same dataset (Down10x dataset), we used a 100 kbp window size with a 50 kbp of step size with custom scripts. We plotted these results together with the results from BCFTools roh on R (version 4.1.2) [42].

The age of coalescence of the ROH tracks was estimated using the rule of g = 100/(2 × RoH_length_) [44], with g being the number of generations. For this, we assumed a constant recombination rate of 1 cM/1 Mbp following [45,46,47].

### 2.10. Past and Present Demographic History

We performed pairwise sequential Markovian coalescent (PSMC) [48] on samples of comparably high coverage (Appendix A). This included hair samples with coverage >7× (three CR hair samples: Afi18, Ken12, Ken18; one WL hair sample: naA2), one CR blood sample (Ggd-Nyango), one GR blood sample (Gbg-Mkubwa), and one MN blood sample (Gbb-Maisha). We used samtools (version 1.11) [30,49] and BCFtools (version 1.6) [30] to call variants and filter for base and mapping qualities <30. We retained sites between a third of the average read depth (-d) and twice the average read depth (-D) for each sample. PSMC was applied using the standard parameters -N25 -t15 -r5 -p “4 + 25 × 2 + 4 + 6”; [48]. We scaled the output using a μ of 1.235 × 10^−8^ and a 19-year generation time, as previously inferred for gorillas by [50].

To study recent demographic changes, we used GONE [51] (Appendix A). GONE relies on linkage disequilibrium (LD), with high power between 0 and 200 generations ago [51]. We obtained PED and MAP files for each subspecies with PLINK [52,53,54] and ran 10 iterations of GONE for each subspecies, where each run had 40 internal replicates. For each run, we used standard input parameters except for the recombination rate of 1.193 cM/Mb [55] and 0.01 as the maximum recombination value. We reduced the maximum hc to avoid bias due to recent migrants and population structure, as suggested by [51].

### 2.11. Assessment of Gene Flow and Connectivity

Using Dsuite [56] on the Down10x dataset, we looked at possible gene flow across gorilla species and geographic areas. We ran Dtrios within the Dsuite package to obtain D statistic and f4-ratio values, which are input for F-branch. We ran the *F*-branch program and its accompanying Python plotting script to generate the matrix displaying excess allele sharing between each branch on the tree and/or its internal branches. First, we ran Dtrios on the species level using the Down10× dataset and one orangutan (Pab-Buschi) from Prado-Martinez et al. [21] as an outgroup. In another iteration, we divided the CR gorilla samples into four groups, each representing a site: Afi, Mbe, Nyango (unknown site), and Kagwene. Finally, to investigate if there are differences in allele sharing across CR sites and WL regions, we grouped several published wild WL gorilla samples into three regions based on their PCA clustering (Appendix A). The regions were the following: the Equatorial Guinea region, with one gorilla for Equatorial Guinea (Ggg-Coco) that clusters with our Monte Alen hair samples; the Naga region, with three samples that cluster around those from Ngaga Camp (Ggg-Carolyn, Ggg-Porta, Ggg-Tzambo), and the Unknown region, including the hair sample of unknown origin (coverage >7×) together with two other previously published samples (Ggg-Akiba-Beri and Ggg-Choomba).

We used ANGSD *D*-statistics [57] to further test for introgression at the site level. Since there is no site information for the published gorilla data, in this case, we restricted this analysis to western gorilla hair samples from Set02 (see Section 2.4, Appendix A). The option -doAbbababa2 [57] allows for multiple individuals per population and was run with the following settings: “angsd -doAbbababa2 1 -bam BAMsList.txt -sizeFile file.size -sites autosomes.file -doCounts 1 -sample 0 -blockSize 5,000,000 -maxDepth 50 -out -anc Hg38.fa -useLast 0 -minQ 20 -minMapQ 30 -p 1”. To estimate *D*-statistics at the species level, we used samples from Set02, in this case, keeping the previously published genomes, with a chimpanzee (Ptt-Clara) from Prado-Martinez et al. [21] as an outgroup. Finally, we also estimated *D*-statistics between WL gorilla regions and GR and between CR and WL gorilla sites. 

To consider admixture at the CR region level (CR-west, CR-central, CR-east), we used an admixture graph approach. We considered five populations: CR-west, CR-central, CR-east, western lowlands, and easterns as an outgroup, using samples from the Down10x dataset. We calculated pairwise f2 statistics for these five populations using the function extract_f2 with blgsize = 500,000 using the R package ADMIXTOOLS 2 [58]. To assess the best-fit topology for a set of admixture events, we used the function find_graphs, with the easterns as the outgroup and varying the number of admixture events from 1 to 10. We assessed which graph had the lowest likelihood score using an out-of-sample score with the function qpgraph. We used the bootstrapping procedure of [58] to assess for significant differences between the graph with the best fitting number of admixture events and the graph with 0 admixture events, using the functions qpgraph_resample_multi with nboot = 100 and compare_fits. We also calculated confidence intervals for the drift lengths and admixture proportions inferred for the graph with 0 admixture events and the graph with the best-inferred number of admixture events using the function qpgraph_resample_snps with boot = 100.

## 3. Results

### 3.1. Single Hairs as a Source for Genomic DNA

We obtained whole-genome data from single hairs collected between 1991 and 2019 from 115 unique individuals, including samples from the poorly known CR subspecies, as well as WL and GR gorillas (Appendix A). The single-hair DNA extracts were sequenced to 30–60 gigabases (Gb) per library (*n* = 116; one sample had two replicate libraries). The average genome-wide coverage reached was 1.35× (min 0.001× and max 13.82×), and the median host DNA (hDNA) was 4.81% (min 0.02% and max 72%) (Appendix A). We identified and excluded five hair samples of chimpanzee origin from our dataset, three of which had less than 1% of human contamination (Appendix A). After quality control (samples with >0.5× coverage and <1% putative human contamination; see Section 2.2), a total of 41 gorilla hair samples were kept (35.3% of the initial dataset), with a mean coverage of 3.23× (median 1.44×) and a mean hDNA of 20.64% (median 14.88%) (Appendix A). This final dataset consists of samples from three gorilla subspecies across eleven sites: four CR gorilla sites (Afi Mountain Wildlife Sanctuary, Afi; Mbe Mountains, Mbe; Cross River National Park Boshi Extension, Bos; and Kagwene Gorilla Sanctuary, Kag), six WL gorilla sites (Deng Deng, Den; Lobéké, Lob; Bai Hokou, Bai; Ngaga Camp, Nga; Monte Alen, MoA; and one unknown, Unk1), and one unknown GR site, Unk3 (Figure 1A, Appendix A). Most of the final 41 samples were identified as female (Appendix A). We identified several first-degree related pairs of gorillas in Afi and Kagwene CR sites that were excluded from population structure analyses (Appendix A, Appendix A).

### 3.2. Increased Resolution of Western Gorilla Population Structure

The four subspecies are clearly separated in the principal component analysis (PCA) (Figure 1B and Appendix A), with hair and blood samples from the same subspecies clustering together. PC1 separates the western and the eastern gorilla species, and PC2 separates the western species into WL and CR subspecies.

When looking specifically at WL gorillas, we observed a main cluster that includes samples from Deng Deng, Lobéké, Bai Hokou, and the unknown site, whereas samples from Monte Alen and Ngaga Camp separate from this cluster in PC1 and PC2, respectively (Figure 1C and Appendix A). This suggests that the sample of unknown geographic origin may belong to a northern WL site. The PCA for CR gorillas shows the Kagwene Gorilla Sanctuary site separated from the rest by PC1 while PC2 separates Afi Mountain Wildlife Sanctuary from CRNP Boshi Extension and Kagwene Gorilla Sanctuary D and Appendix A). As expected, F_ST_ analysis revealed that sampling sites from the same subspecies have higher genetic similarity than when comparing sites between the western gorilla subspecies (WL vs. CR) (Figure 1E, Appendix A). Among CR gorilla sites, Kag (Kagwene Gorilla Sanctuary) is the most genetically differentiated using F_ST_ (Figure 1E, Appendix A), supporting the PCA clustering and its separation along PC1 (Figure 1D).

Given that most hair samples included in the study have known GPS coordinates, we were able to better determine the putative geographic origin of the previously published high-coverage samples, for which only the area or country of origin are available as metadata (see the supplementary table in Prado-Martinez et al. [21] and Xue et al. [31]). For example, the only previously published whole genome sample of a CR gorilla, Nyango, clusters near the Mbe and Boshi sites (Appendix A). In the instance of WL, we were able to confirm that Ggg-Coco is from Equatorial Guinea, as observed from its clustering near the Monte Alen samples (Appendix A).

### 3.3. Genetic Diversity among Gorilla Species

Genome-wide heterozygosity estimates on hair samples show that Cross River gorillas present lower heterozygosity levels than WL but higher levels than both eastern gorilla subspecies (Appendix A). We observe little variation in heterozygosity levels among CR sites. Monte Alen presents the highest heterozygosity among WL sites, and there is a high variance in heterozygosity among WL sites, in contrast to the trend among CR sites. The overall fraction of the genome in runs of homozygosity (ROH) is higher in CR gorillas than in WL gorillas but lower compared to the eastern gorilla species. Nonetheless, the fraction of the genome in long ROH, which is a proxy of recent inbreeding levels, showed that CR gorillas have higher levels of recent inbreeding than Grauer’s gorillas but lower than mountain gorillas (Figure 2A, Appendix A).

### 3.4. Strong Bottleneck in Cross River Gorillas

PSMC results of the CR gorilla hair samples with higher coverage (Appendix A) recapitulate the trends for changes in effective population size (Ne) inferred using the single CR blood sample (Ggd-Nyango) (Appendix A). The trajectories in Ne for these four CR individuals are highly concordant, except in very recent times where the method has reduced power. This is in agreement with recent population split times of CR gorilla populations. To assess population demography in more recent times, we implemented GONE, which revealed a strong recent bottleneck in CR gorillas (Figure 2B). In the last 10 generations, we infer the Ne of CR gorillas was more similar to that of MN than to GR gorillas. Considering ROH of length 5–10 Mb to calculate the age of these ROH tracks, we likewise infer an onset of inbreeding 5–10 generations ago, which would correspond to around 100–200 years ago. These results indicate recent inbreeding, similar to what is known for MN. 

### 3.5. Past Gene Flow among Gorilla Subspecies

On the sharing of alleles among western gorillas, between CR sites and WL sites, with D statistics, we see the highest allele sharing between the northern WL site cluster (Bai-Hokou, Deng Deng, and Lobéké) and all CR sites excluding the CRNP Boshi Extension site (Figure 3A, Appendix A). To assess this signal at a higher resolution, we implemented the F-branch method, which takes into account the correlation among different branches of the tree. As such, we were able to infer that the common ancestor of the northern WL cluster had gene flow with CR gorillas (Appendix A).

Within CR sites, we infer the highest levels of allele sharing between Afi Mountain and Mbe, which are geographically closely located (Figure 3B and Appendix A). Furthermore, using the F-branch method and Dstats, we uncover a signal of past gene flow between CR gorillas and the eastern GR gorilla subspecies (Figure 3B and Appendix A). Taking our gene flow analyses together, the CR CRNP Boshi Extension and the WL Monte Alen sites show weaker allele-sharing signals compared to other sites (Figure 3A and Appendix A).

To investigate admixture within CR regions (CR-west, Afi Mountain Wildlife Sanctuary site; CR-central, Mbe Mountains site and Nyango sample; and CR-eastern, Kagwene Gorilla Sanctuary site), we constructed admixture graphs of varying numbers of admixture events (see Section 2.11). The best-fitting admixture graph (likelihood = 0.001712409) infers high shared ancestry between western lowlands and CR-east (Kagwene Gorilla Sanctuary) and models CR-west (Afi Mountain Wildlife Sanctuary) and CR-central (Mbe Mountains site and Nyango) populations as admixed with a major CR-east (Kagwene Gorilla Sanctuary) component and a more basal, minor component (Appendix A). Despite having a much lower likelihood than a model with zero admixture events (likelihood = 100.9402) (Appendix A), when considering SNP variance by bootstrapping, these two models are not significantly different (*p* = 0.4970), likely because they share topological features. Nonetheless, we consider the best-fitting admixture graph to be a reasonable model for this group of populations since the CR-east (Kagwene Gorilla Sanctuary) population lives in closer proximity to western lowland gorilla populations than the two other regions, CR-west and CR-central. In principle, such allele sharing could be the consequence of admixture at some point in the past or due to ancient substructures within the CR gorillas. However, given the very similar population size trajectories of CR gorillas, likely reflecting recent split times, the time depth might not be sufficient for such an ancient substructure to occur.

## 4. Discussion

In this study, we have considerably expanded the number of whole genomes available for CR gorillas, from only 1 whole genome previously available [21] to 25 new samples that pass our stringent QC. Together with the new 15 WL gorilla genomes, we provide the most comprehensive whole-genome dataset of wild western gorillas to date, including publicly available genomes derived from invasive samples [21,31]. As such, we have been able to explore at greater resolution genome-wide heterozygosity, inbreeding, demography, and connectivity of CR gorillas, the subspecies for which population-level data had been missing.

Previous studies on great apes using non-invasive samples have been restricted to analyses of neutral markers, e.g., microsatellites [18,20,59,60,61,62], and more recently target enrichment of specific genomic regions, e.g., chromosome 21 [45]. To date, only one study has obtained WGS data using multiple hairs from a wild-threatened species [63]. In this study, we have used a single hair per extraction and proven the feasibility of obtaining reliable whole-genomic data for population genomic studies from this type of sample. 

Using our new genomes with the publicly available wild gorilla genomes, we observe for the first time four differentiated clusters using the first and second principal components of the PCA, which represent the four gorilla species and subspecies. This indicates that although we included different sample types (blood and hair) from different studies, we do not observe major technical effects, and the clustering seen here reflects biological differences.

Our genome-wide heterozygosity estimates are in line with previous studies, in which CR gorillas showed lower heterozygosity levels than WL gorillas but higher heterozygosity than the two eastern subspecies, GR and MN gorillas [14,18,21,31,46,61,64]. However, we also uncovered for the first time long ROH (>10 Mb) in CR gorillas, which is indicative of recent inbreeding caused by the mating of related individuals in such small populations. In fact, we dated the age of ROH tracks >5–10 Mb to around 100–200 ya. This matches the timing of a strong recent bottleneck in CR gorillas inferred in our GONE analyses. In fact, CR gorillas appear to have suffered an even more recent bottleneck than both eastern gorilla subspecies. Hence, while MN gorillas have experienced long-term low effective population size [46], our findings indicate that CR gorillas are currently at greater risk of a higher realized genetic load [65,66,67]. This finding is in agreement with previous studies that observed evidence of a recent and abrupt reduction in CR gorilla population size [14,22], probably caused by hunting in the past two hundred years, as suggested by Bergl et al. [14].

Previous studies using microsatellites on CR gorilla fecal samples found a population structure in CR gorillas consistent with geography and habitat fragmentation [14,19]. As such, three main CR subpopulations were defined: western, comprising Afi Mountain Wildlife Sanctuary; eastern, comprising Kagwene Gorilla Sanctuary; and central, comprising the remaining study sites (Mbe Mountains, Mone, Takamanda south, and Boshi). In this study, we included samples from these three regions, and our results using WGS support those obtained with microsatellite data.

Our analyses revealed past gene flow among two nearby CR sites and suggested the presence of an isolated CR site. Specifically, the observed gene flow between Afi Mountain and Mbe, which are geographically close, suggests that these two communities were connected in the past but have been subjected to high levels of habitat fragmentation and human disturbance over time. This confirms previous findings by Imong et al. [12]. On the other hand, the CRNP Boshi Extension site showed the lowest gene flow signal, confirming the reduced functional connectivity between this and other CR sites previously observed [3]. Nonetheless, obtaining samples from closer sites, such as CRNP Okwa Hills, could help increase the resolution of these results. 

Additionally, we observed past gene flow between CR sites and the northern WL sites. This agrees with findings from mitochondrial haplogroups observed by Clifford et al. [16] and Anthony et al. [17], where one haplogroup extended from the CR area in Nigeria to southeastern Cameroon (part of WL distribution). This signal of gene flow between the two western subspecies is consistent with microsatellite results using feces and 100-year-old teeth roots [22]. However, this signal was not observable using only one WGS CR gorilla sample, Nyango [23], demonstrating the importance of population-level WGS data for such analyses. Moreover, Fontsere et al. [45] recently observed gene flow between a chimpanzee site located nearby Lobéké (WL gorilla site) and chimpanzees located in the northern distribution of CR gorillas. Taken together, these results indicate the existence of a connectivity corridor in the past used by both chimpanzees and gorillas. 

Furthermore, we detected a signal of past gene flow between CR gorillas and eastern gorillas, with a stronger signal with GR than MN gorillas, consistent with previous studies [20,68]. This finding was not detected in a previous analysis using Nyango, the single WGS CR individual to date [23]. The increased number of CR samples in the current study improves the resolution of inferences. Hence, all the evidence points to a complex demographic history of CR gorillas with several episodes of gene flow, which would require future in-depth modeling to better understand their demography. 

We performed thorough quality control, resulting in a filtered dataset that represents 35% of the original set of samples, highlighting the low quality and amount of data retrieved from shed hair samples. This was to be expected, given that most of the single shed hairs we used were collected in the early 1990s from gorilla nests. As such, the samples from the CRNP Okwa Hill, Takamanda National Park and Mt. Doudou sites were not of sufficient quality for further analysis. We stress the importance of quality control when working with non-invasive samples because most of the samples from the Takamanda National Park site belonged to chimpanzees instead of gorillas.

In this study, we have focused on the two subspecies of western gorillas and complemented previously published blood samples with newly sequenced WG data from single hairs. Additional samples from geolocalized eastern gorilla sites would allow future studies to further investigate past connectivity in the gorilla species. Additionally, increasing the number of geolocalized samples for the CR sites would help gain resolution on differences among regions or sites. Furthermore, studying which CR introgressed tracks are derived from WL or GR gorillas and the time of these admixture events could help characterize some functional consequences of these introgressed fragments. 

We demonstrate that Cross River gorillas display recent inbreeding and a population bottleneck despite signals of past connectivity with other gorillas and gene flow within the Cross River population. These findings are almost certainly associated with the gorillas’ current concentration in small, restricted, hilly areas, which hinders genetic connectivity. If current conditions persist, ongoing inbreeding and the associated loss of genetic diversity would very likely reduce the fitness of the population and increase their extinction risk. However, our data also reinforce previous findings [14] that since the bottleneck in the Cross River population is relatively recent, further loss of genetic diversity could be minimized if the population is allowed to expand. Since large areas of unoccupied apparent gorilla habitat remain in the Cross River landscape [3,12], and camera trap images suggest the population is reproducing (WCS unpublished data), it is possible that population growth could be achieved if hunting pressure were relaxed. Our study also illustrates that single hairs are a viable source of genomic information. Future genomic studies aiming to help the conservation of threatened species will undoubtedly benefit from this non-invasive sample type.

## Figures and Tables

**Figure 1 genes-14-00743-f001:**
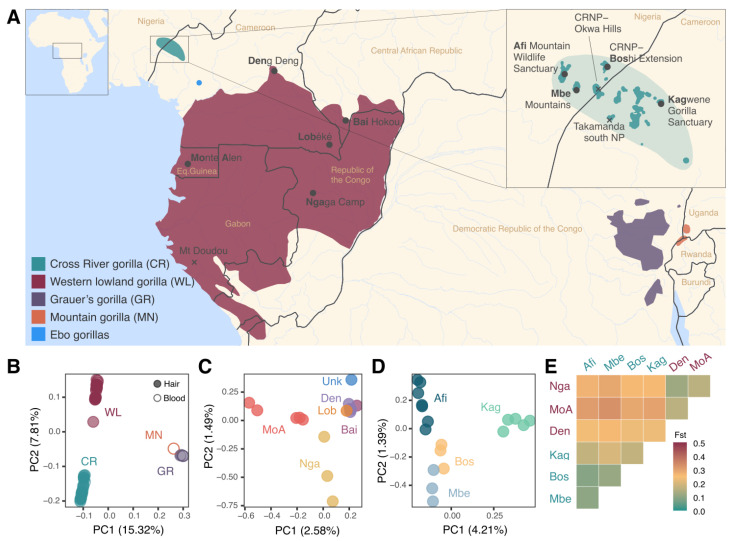
(**A**) Map showing the geographic distribution of the four gorilla subspecies. The Cross River and MN gorilla distribution ranges are from [2], and the other subspecies shape files were obtained from the IUCN. The shape files showing the specific Cross River gorilla populations (darker green) have been adapted with permission from [2] (Supplementary File). Inset at the top right of the map shows the Cross River distribution in more detail. Black dots indicate sites for which we have samples in the final dataset. Black crosses indicate sampled sites for which samples were discarded after quality control (Takamanda NP (National Park), CRNP Okwa Hills (Cross River National Park, Okwa Hills), and Mt. Doudou). Samples of unknown geographic origin are not represented in this map. (**B**) Principal component analysis (PCA) including hair samples (filled circles) and blood samples (empty circles) of the four subspecies. (**C**) PCA of western lowland gorilla hair samples with color indicating each site (Monte Alen, MoA; Ngaga Camp, Nga; Deng Deng, Den; Bai Hokou, Bai; unknown origin, Unk1). (**D**) PCA of Cross River gorilla hair samples with color indicating each site (Afi Mountain Wildlife Sanctuary, Afi; CRNP Boshi Extension, Bos; Mbe Mountains, Mbe; and Kagwene Gorilla Sanctuary, Kag). (**E**) Heatmap of pairwise F_ST_ of hair samples for both western gorilla species, CR and WL. Site labels are colored to indicate species, with red for WL gorilla sites and green for CR gorilla sites. Cell color indicates Fst value.

**Figure 2 genes-14-00743-f002:**
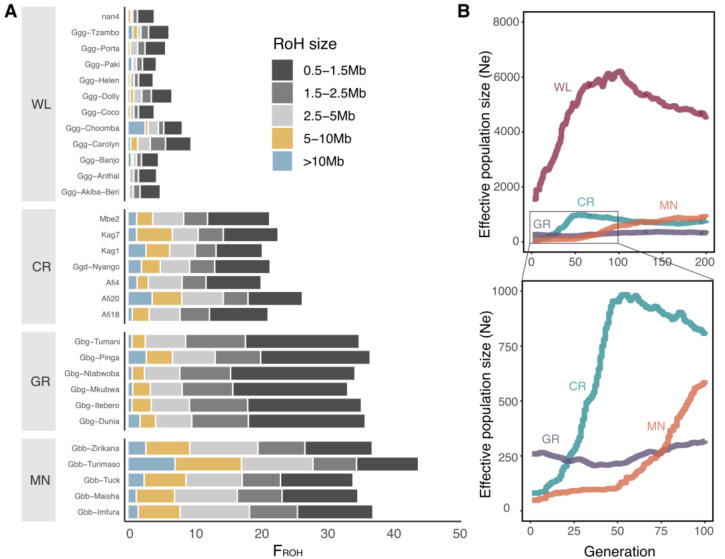
(**A**) FROH (percentage of the genome in runs of homozygosity (ROH) for different ROH size ranges (0.5–1.5 Mb, dark grey; 1.5–2.5 Mb, grey; 2.5–5 Mb, light grey; 5–10 Mb, yellow; and >10 Mb, blue). (**B**) Inference of recent effective population size from GONE for each subspecies (Cross River gorilla, CR; western lowland gorilla, WL; Grauer’s gorilla, GR and mountain gorilla, MN).

**Figure 3 genes-14-00743-f003:**
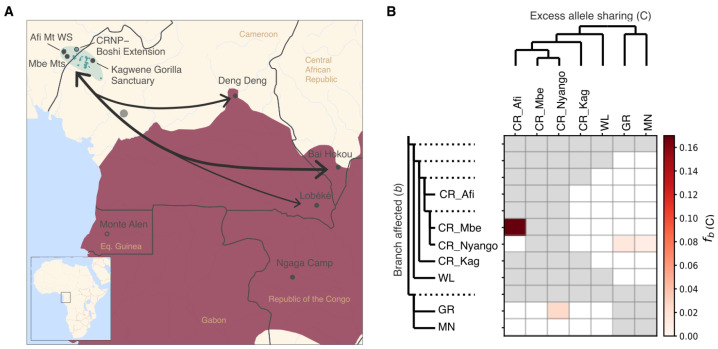
(**A**) Representation of the main results from D-statistics analysis between Cross River (CR) sites and western lowland (WL) gorilla regions. Arrows indicate which western lowland (WL) sites have higher allele sharing with Cross River (CR) gorillas. Arrow thickness indicates the level of gene flow, with the thicker arrow indicating higher gene flow between WL and CR sites. Empty dots indicate sites (CRNP Boshi Extension and Monte Alen) that have no positive signal of allele sharing with the other gorilla subspecies. (**B**) F-branch (fb(C)) analysis on excess allele sharing between species/populations/regions. The *x*-axis and *y*-axis trees are the species tree and the expanded species trees (with internal branches indicated with dotted lines), respectively. Grey-colored cells indicate comparisons for which gene flow could not be inferred based on the given tree topology. Red color indicates fb value; the darker color indicates greater allele sharing between the expanded tree branch (relative to its sister branch) (b) and the populations on the *x*-axis (C).

## Data Availability

Genome resequencing data are freely available in ENA under the project number (PRJEB60463).

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
