# Peer review of "Past Connectivity but Recent Inbreeding in Cross River Gorillas Determined Using Whole Genomes from Single Hairs"

_genes, 2023, doi:10.3390/genes14030743_

Round 1

Reviewer 1 Report

The paper deals with the past connectivity but recent inbreeding in Cross River gorillas  determined using whole genomes from single hairs. The topic of the paper is current and important because the estimation of genetic variability within populations is important whereas it is related to the evolutionary potential of populations, which is usually higher in populations with greater genetic variation. Introduction and methods are well written. The results of the obtained research were properly described and discussed and brings some new knowledge.

Author Response

Dear Reviewer 1, 

We wanted to express our gratitude for taking the time to review our manuscript entitled "Past connectivity but recent inbreeding in Cross River gorillas determined using whole genomes from single hairs".

We are pleased to read that you find that our work brings new knowledge and meets the standards to be published to this special issue. 

Thank you once again for your valuable feedback and your time and effort in reviewing our manuscript. 

Best regards,

Marina Alvarez-Estape & Tomas Marques-Bonet (and the rest of the co-authors)

Reviewer 2 Report

Kindly correct as suggested

Round 2

Reviewer 2 Report

Dear Sir

The article has been corrected as per the suggestions and may be accepted for publication

kr

akt

Author Response

Dear Reviewer, 

We are very thankful for your time and effort to review our manuscript and our comments on your suggestions.

We are pleased to read that you are satisfied with the modifications and that you find our version of the manuscript suitable for publication.

Thank you again for your valuable feedback.

Best regards,

Marina Alvarez-Estape and Tomas Marques-Bonet